# Incorporating Inductive Biases to Energy-based Generative Models

**Yukun Li**                                                                          *yukun.li@tufts.edu*
*Department of Computer Science*
*Tufts University*

**Li-Ping Liu**                                                                      *liping.liu@tufts.edu*
*Department of Computer Science*
*Tufts University*

**Reviewed on OpenReview:** *https://openreview.net/forum?id=k98ZDblyhN*

## Abstract

With the advent of score-matching techniques for model training and Langevin dynamics for sample generation, energy-based models (EBMs) have gained renewed interest as generative models. Recent EBMs usually use neural networks to define their energy functions. In this work, we introduce a novel hybrid approach that combines an EBM with an exponential family model to incorporate inductive bias into data modeling. Specifically, we augment the energy term with a parameter-free statistic function to help the model capture key data statistics. Like an exponential family model, the hybrid model aims to align the distribution statistics with data statistics during model training, even when it only approximately maximizes the data likelihood. This property enables us to impose constraints on the hybrid model. Our empirical study validates the hybrid model's ability to match statistics. Furthermore, experimental results show that data fitting and generation improve when suitable informative statistics are incorporated into the hybrid model.

## 1 Introduction

Energy-based models (EBMs) (Murphy, 2012; LeCun et al., 2006; Ngiam et al., 2011), which provide a flexible way of parameterization, are widely used in data modeling such as sensitive estimation (Nguyen & Reiter, 2015; Wenliang et al., 2019; Jiang & Xiao, 2021), structured prediction (Belanger & McCallum, 2016; He & Jiang, 2020; Pan et al., 2020), and anomaly detection (Zhai et al., 2016; Liu et al., 2020). They recently have achieved successes as generative models for data of different modalities, such as images (Du & Mordatch, 2019; Vahdat & Kautz, 2020), texts (Deng et al., 2020; Yu et al., 2022), graph (Liu et al., 2021; Chen et al., 2022; Xu et al., 2022) and point cloud (Xie et al., 2020; 2021; Luo & Hu, 2021), thanks to the advent of new training methods of score matching (Song & Ermon, 2019; Song et al., 2020b; Ho et al., 2020).

Previous EBMs often have linear energy functions in model parameters, so they fall into the category of exponential-family models (Wainwright et al., 2008). An exponential family model has a few nice properties. It matches data statics when it maximizes the likelihood of the training data. Among all models, it also has the largest entropy with the constraint of matching statistics. When constructing an EBM, statistic functions are often used to capture key statistics of the data. However, in recent years, EBMs have used neural networks as their energy functions. Since their energy functions are not linear in trainable parameters, they are not exponential-family models anymore.

In our construction of EBMs, we propose to include linear terms defined with special statistic functions in the energy function to retrieve some good properties of exponential-family models. Even with complex data types such as point clouds and graphs, we still often have domain knowledge to specify statistic functions so that the model can capture desired properties. Therefore, the linear term with special statistic functions

allows us to inject inductive biases in model fitting. This framework applies to the domains where prior information can be written in statistic functions.

The proposed method is very useful for data fitting in multiple iterations. Note that data fitting is often an iterative procedure (Gelman et al., 1995), and the goodness of fit is often monitored by the discrepancy between statistics computed respectively from model samples and real data. With our method, we can improve the model by adding corresponding terms to narrow the discrepancy. Taking molecule generation as an example, when we observe molecule samples violate the valency constraint, we can incorporate the knowledge of valency rules to improve the model.

We evaluate the effectiveness of this new approach on three tasks: modeling molecular data, fitting handwritten digits, and modeling point clouds. The results demonstrate the effectiveness of the proposed approach in a wide range of data-fitting domains.

## 2 Related Work

The need to incorporate inductive bias into a generative model has drawn researchers' attention for a long time. Zhu et al. (1997) proposes the general principle for incorporating prior knowledge into a generative model. Khalifa et al. (2020) introduces a novel method for text generation using EBMs to enforce desired statistical constraints. Their framework employs the EBM model to approximate a pre-trained language model instead of directly fitting the data. Korbak et al. (2021) extends this approach to code generation, imposing a constraint that generated sequences are compilable. Qin et al. (2022) applies energy constraints to a transformer decoder to control text semantics and style. Lafon et al. (2023) learns the hybrid model combined with the Gaussian Mixture Model (GMM) and EBMs for out-of-distribution detection. In contrast, our work focuses on improving data fitting with inductive bias.

Another thread of research changes a generative model's behavior during the sampling stage (Dhariwal & Nichol, 2021). For example, Chung et al. (2022) proposes to use manifold constraint to guide a pre-trained diffusion model's sampling path. Kim et al. (2022); Zhao et al. (2022); Song et al. (2022); Yu et al. (2023); Liu et al. (2023); Bansal et al. (2023); Li & Liu (2024) propose different guidance for different specific problems to boost the performance of each task. In contrast, our method aims to fit the original data distribution more accurately by adding the statistics term. Notably, the classifier-guidance method operates during inference with manually tuned weights, whereas our model learns the inductive bias strength corresponding to data distribution statistics.

## 3 Background

### 3.1 Score-matching for energy-based models

An energy-based model is defined with an energy function $E_{\boldsymbol{\theta}}$ parameterized by $\boldsymbol{\theta}$,

$$p_{\boldsymbol{\theta}}(\mathbf{x}) = \frac{\exp(-E_{\boldsymbol{\theta}}(\mathbf{x}))}{Z_{\boldsymbol{\theta}}}. \tag{1}$$

Here, $\mathbf{x} \in \mathbf{R}^d$, d is the feature dimension. $Z_{\boldsymbol{\theta}}$ is the partition function: $Z_{\boldsymbol{\theta}} = \int \exp(-E_{\boldsymbol{\theta}}(\mathbf{x}))d\mathbf{x}$, which is usually intractable. Various methods exist to train an energy-based model. Maximum likelihood training with Markov Chain Monte Carlo (MCMC) sampling (Hinton, 2002) is one standard method to train the Energy-based model, but MCMC is typically computationally expensive.

A modern EBM often devises the energy function $E_{\boldsymbol{\theta}}(\mathbf{x})$ with a neural network, and $\boldsymbol{\theta}$ represents learnable parameters of the neural network (Song & Kingma, 2021). These models are usually fit by score matching (Hyvärinen & Dayan, 2005; Kingma & Cun, 2010), which avoids the explicit calculation of the $Z_{\boldsymbol{\theta}}$. Score matching minimizes the mean square error between the model score function $\mathbf{s}_{\boldsymbol{\theta}}(\mathbf{x}) = \nabla_{\mathbf{x}} \log p_{\boldsymbol{\theta}}(\mathbf{x})$ and the data score function $\nabla_{\mathbf{x}} \log p_{data}(\mathbf{x})$:

$$D_F(p_{data}(\mathbf{x})||p_{\boldsymbol{\theta}}(\mathbf{x})) = E_{p_{data}(\mathbf{x})}\left[\frac{1}{2}||\nabla_{\mathbf{x}} \log p_{data}(\mathbf{x}) - \nabla_{\mathbf{x}} \log p_{\boldsymbol{\theta}}(\mathbf{x})||_2^2\right]. \tag{2}$$

where $p_{data}(\mathbf{x})$ is the true data density function, $p_{\boldsymbol{\theta}}(\mathbf{x})$ is the model density function. To bypass the intractable of the first term, we often use integration by parts to rewrite it as :

$$D_F(p_{data}(\mathbf{x})||p_{\boldsymbol{\theta}}(\mathbf{x})) = E_{p_{data}(\mathbf{x})} \left[ \frac{1}{2} \sum_{i=1}^{d} \left( \frac{\partial E_{\boldsymbol{\theta}}(\mathbf{x})}{\partial x^i} \right)^2 + \frac{\partial^2 E_{\boldsymbol{\theta}}(\mathbf{x})}{(\partial x^i)^2} \right] + const. \tag{3}$$

Several efficient variants of the score matching have been proposed. One is Sliced Score Matching (Song et al., 2020a), which randomly samples a projection vector $\mathbf{v}$, takes the inner product between $\mathbf{v}$ and the two scores, and then compares the resulting two scalars:

$$D_F(p_{data}(\mathbf{x})||p_{\boldsymbol{\theta}}(\mathbf{x})) = E_{p_{data}(\mathbf{x})} E_{p(\mathbf{v})} \left[ \frac{1}{2} (\mathbf{v}^\top \nabla_{\mathbf{x}} \log p_{data}(\mathbf{x}) - \mathbf{v}^\top \nabla_{\mathbf{x}} \log p_{\boldsymbol{\theta}}(\mathbf{x}))^2 \right]. \tag{4}$$

The parallelizable variant of sliced Score Matching is Finite difference sliced score matching (FDSSM) (Pang et al., 2020). Another score-matching variant is Denoising Score Matching (Vincent, 2011)(DSM). DSM avoids the computation of the second-order derivatives by adding some small noise perturbation to the data. Then DSM learns the noisy data distributions $q(\tilde{\mathbf{x}}) \approx p_{data}(\mathbf{x})$:

$$D_F(q(\tilde{\mathbf{x}})||p_{\boldsymbol{\theta}}(\tilde{\mathbf{x}})) = E_{q(\mathbf{x},\tilde{\mathbf{x}})} \left[ \frac{1}{2} ||\nabla_{\mathbf{x}} \log q(\tilde{\mathbf{x}}|\mathbf{x}) - \nabla_{\mathbf{x}} \log p_{\boldsymbol{\theta}}(\tilde{\mathbf{x}})||_2^2 \right]. \tag{5}$$

## 3.2 Exponential-family distributions

A distribution from the exponential family can be viewed as a special type of EBM whose energy function is linear in its parameters. Various common distributions such as Gaussian, Bernoulli, and Poisson are in Exponential-family distributions. Here we consider continuous distributions whose probability density function (PDF) takes the following form:

$$p(\mathbf{x}; \boldsymbol{\eta}) = \frac{\exp(\boldsymbol{\eta}^\top \mathbf{T}(\mathbf{x}))}{Z(\boldsymbol{\eta})}. \tag{6}$$

In this form, we assume the base measure is 1. The statistic function $\mathbf{T}(\mathbf{x})$ decides the distribution type, while the parameter $\boldsymbol{\eta}$ decides the actual distribution of that type. The partition function $Z(\boldsymbol{\eta})$, which is often hard to compute, ensures that the PDF integrates to 1 over the domain of the distribution.

The statistic function $\mathbf{T}(\mathbf{x})$ is also meaningful in capturing data statistics. The statistics $\frac{1}{N} \sum_{i=1}^{N} \mathbf{T}(\mathbf{x}_i)$ computed from a dataset $(\mathbf{x}_i : i = 1, \dots, N)$ is called the data's *sufficient statistics*, which provide sufficient information about the distribution parameter. Data fitting with $p(\mathbf{x}; \boldsymbol{\eta})$ is centering at sufficient statistics: if the data likelihood is maximized with a particular $\boldsymbol{\eta}$ under the model $p(\mathbf{x}; \boldsymbol{\eta})$, then $\mathbb{E}_{p(\mathbf{x};\boldsymbol{\eta})}[\mathbf{T}(\mathbf{x})] = \frac{1}{N} \sum_{i=1}^{N} \mathbf{T}(\mathbf{x}_i)$.

## 4 A Hybrid Energy-based Model with Inductive Bias

This work proposes combining neural energy functions and interpretable statistic functions to develop EBMs. The hybrid model takes the following form:

$$p_{\boldsymbol{\theta}, \boldsymbol{\eta}}(\mathbf{x}) = \frac{\exp \left( F_{\boldsymbol{\theta}}(\mathbf{x}) + \boldsymbol{\eta}^\top \mathbf{T}(\mathbf{x}) \right)}{Z(\boldsymbol{\theta}, \boldsymbol{\eta})}. \tag{7}$$

Here $Z(\boldsymbol{\theta}, \boldsymbol{\eta}) = \int_{\mathbf{x}} \exp \left( F_{\boldsymbol{\theta}}(\mathbf{x}) + \boldsymbol{\eta}^\top \mathbf{T}(\mathbf{x}) \right) d\mathbf{x}$ is the partition function. $F_{\boldsymbol{\theta}}(\mathbf{x}) \in \mathbf{R}$ is a real-valued function constructed with a neural network parameterized by $\boldsymbol{\theta}$. The statistic function $\mathbf{T}(\mathbf{x})$ itself has no learnable parameters, but it is multiplied to the learnable parameter $\boldsymbol{\eta}$ in the linear term.

The hybrid model still possesses a nice property of the exponential family distribution. Model fitting still aims to match the distribution mean of sufficient statistics to the sample mean (Wainwright et al., 2008). Let $l(\boldsymbol{\theta}, \boldsymbol{\eta})$ denote the log-likelihood of the data, $A(\boldsymbol{\theta}, \boldsymbol{\eta}) = \log Z(\boldsymbol{\theta}, \boldsymbol{\eta})$:

$$l(\boldsymbol{\theta}, \boldsymbol{\eta}) = \sum_{i=1}^{n} F_{\boldsymbol{\theta}}(\mathbf{x}_i) + \sum_{i=1}^{N} \boldsymbol{\eta}^\top \mathbf{T}(\mathbf{x}_i) - N A(\boldsymbol{\theta}, \boldsymbol{\eta}). \tag{8}$$

**Theorem 1.** *If the parameter $\boldsymbol{\eta}$ is at a local maximum of the $l(\boldsymbol{\theta}, \boldsymbol{\eta})$ for a fixed $\boldsymbol{\theta}$, then $\mathbb{E}_{p_{\boldsymbol{\theta}, \boldsymbol{\eta}}}\left[\mathbf{T}(\mathbf{x})\right] = \frac{1}{N}\sum_{i=1}^{N}\mathbf{T}(\mathbf{x}_i)$.*

*Proof.* If we treat $\exp\left(F_{\boldsymbol{\theta}}(\mathbf{x})\right)$ as a base measure, then $p_{\boldsymbol{\theta}, \boldsymbol{\eta}}(\mathbf{x})$ is an exponential-family distribution. Then, the gradient of data likelihood with respect $\boldsymbol{\eta}$ can be derived according to Wainwright et al. (2008) (eq. 3.38 and Prop. 3.1). By taking the derivative of $L(\boldsymbol{\theta}, \boldsymbol{\eta})$ with respect to $\boldsymbol{\eta}$, we have

$$\nabla_{\boldsymbol{\eta}}l = \sum_{i=1}^{N}\mathbf{T}(\mathbf{x}_i) - N \cdot \mathbb{E}_{p_{\boldsymbol{\theta}, \boldsymbol{\eta}}}[\mathbf{T}(\mathbf{x})] \tag{9}$$

If the weight is at the local minimum, then $\nabla_{\eta}l = 0$, which leads to the conclusion. $\square$

With this property, we will construct a hybrid model with specific statistics in its linear energy terms. Model fitting will pay attention to the specified statistics in the data. It can be viewed as an approach to injecting inductive bias into the model. Before we discuss such models in real applications, we first consider their training procedure.

## 4.1 Training the model

We choose the score matching method to train the EBMs given its efficiency and effectiveness Song et al. (2020b). The score function for this hybrid model is:

$$\mathbf{s}_{\boldsymbol{\theta}, \boldsymbol{\eta}}(\mathbf{x}) = \nabla_{\mathbf{x}}F_{\boldsymbol{\theta}}(\mathbf{x}) + \nabla_{\mathbf{x}}\mathbf{T}(\mathbf{x}) \cdot \boldsymbol{\eta} \tag{10}$$

We can either take derivative of $F_{\theta}(\mathbf{x})$ to get $\nabla_{\mathbf{x}}F_{\boldsymbol{\theta}}(\mathbf{x})$, or directly specify $\nabla_{\mathbf{x}}F_{\boldsymbol{\theta}}(\mathbf{x})$ without specifying $F_{\theta}(\mathbf{x})$. In the latter case, $\nabla_{\mathbf{x}}F_{\boldsymbol{\theta}}(\mathbf{x})$ is approximated by a neural network with parameter $\theta$. Once we are here, we follow the standard procedure of denoising score matching (Vincent, 2011) to train our model:

$$\mathbb{E}_{\mathbf{x}\sim p(\mathbf{x})}\mathbb{E}_{\tilde{\mathbf{x}}\sim q_{\sigma}(\tilde{\mathbf{x}}|\mathbf{x})}[||\nabla_{\mathbf{x}}\log q_{\sigma}(\tilde{\mathbf{x}}|\mathbf{x}) - \mathbf{s}_{\boldsymbol{\theta}, \boldsymbol{\eta}}(\mathbf{x})||_2^2]. \tag{11}$$

Where $\tilde{\mathbf{x}}$ is the perturbed data. To learn the score function better in the low-density regions, we adopt the technique by perturbing the data with K different noise levels $\{\sigma_i\}_{i=1}^K$ and learn the noise-conditioned score network (Song & Ermon, 2019) with the following loss:

$$\mathcal{L}(\boldsymbol{\theta}, \boldsymbol{\eta}; \{\sigma_i\}_{i=1}^K) = \frac{1}{K}\sum_{i=1}^{K}\lambda(\sigma_i)\mathbb{E}_{\mathbf{x}\sim p_{data}(\mathbf{x})}\mathbb{E}_{\tilde{\mathbf{x}}\sim q_{\sigma_i}(\tilde{\mathbf{x}}|\mathbf{x})}[||\nabla_{\mathbf{x}}\log q_{\sigma_i}(\tilde{\mathbf{x}}|\mathbf{x}) - \mathbf{s}_{\boldsymbol{\theta}, \boldsymbol{\eta}}(\mathbf{x})||_2^2]. \tag{12}$$

Here $\lambda(\sigma_i)$ is the weight for each noise level.

Given the approximation here, one question is whether the trained model still matches the data statistics. Note that the score-matching loss is the lower bound of the log-likelihood (Song et al., 2021) with appropriate weighting $\lambda$.

$$\sum_{i=1}^{n}\log p_{\boldsymbol{\theta}, \boldsymbol{\eta}}(\mathbf{x}_i) \geq L(\boldsymbol{\theta}, \boldsymbol{\eta}; \lambda) \tag{13}$$

Model training with score matching approximately maximizes the data likelihood. With a similar spirit, the learned model approximately matches the data statistics. Furthermore, the difference between the distribution mean and the data mean: $\varDelta_{\mathbf{T}(\mathbf{x})} = \mathbb{E}_{\mathbf{x}\sim p_{data}}[\mathbf{T_x}] - \mathbb{E}_{\mathbf{x}\sim p_{model}}[\mathbf{T_x}]$ approaches to zero as the model parameter $\boldsymbol{\eta}$ approaches a local maximum of the log-likelihood. This can be proved based on the convexity of the log-likelihood function w.r.t. parameter $\boldsymbol{\eta}$. For any $\theta$ fixed, $\exp\left(F_{\boldsymbol{\theta}}(\mathbf{x})\right)$ can be viewed as a base measure, then the hybrid distribution in (7) falls in the exponential family of distributions, and its log-likelihood is convex w.r.t. $\boldsymbol{\eta}$ by Wainwright et al. (2008). Therefore, parameters $(\boldsymbol{\theta}^*, \boldsymbol{\eta}^*)$ at a local maximum of the likelihood indicates that $\boldsymbol{\eta}^*$ is at the maximum of the likelihood function with $\boldsymbol{\theta}^*$ fixed. From convex analysis (Boyd & Vandenberghe, 2004), the norm of the gradient approaches zero as $(\boldsymbol{\theta}, \boldsymbol{\eta})$ approaches the local maximum.

### 4.2  Inject inductive bias through the function $\mathbf{T}(\mathbf{x})$

As in an exponential-family model, the statistic function points the hybrid model's attention to special statistics of data distribution. Therefore, it is a convenient approach to inject inductive bias into the model. Without any restrictions over the function form, the statistic function is a convenient approach to express such inductive bias. In this section, we leverage this property to develop models for three applications to show its practical value.

#### 4.2.1  Statistic function for fitting molecule data

A molecular generative model is an important tool for chemical applications, and it is often defined as an EBM (Liu et al., 2021; Niu et al., 2020; Jo et al., 2022). Here, we consider two molecular generative models, EDP-GNN Niu et al. (2020) and GDSS Jo et al. (2022), which both can be viewed as energy-based models and specify distributions of molecules through their molecular graphs and atom types. Assume the atom has $k$ types. In these two models, the random variable $\mathbf{x} = (\mathbf{b}, \mathbf{A})$ represents a molecule graph containing $n$ nodes (atoms), with $\mathbf{b} \in \{1, \ldots, k\}^n$ representing atom types in a molecule, and $\mathbf{A}$ being the adjacency matrix representing the connectivity of corresponding atoms. Without the statistic function $T$, these two models use a neural network to define $\nabla_{\mathbf{x}} F_{\boldsymbol{\theta}}(\mathbf{x})$.

An energy-based model based on a neural function often assigns non-zero probabilities to "invalid" molecules even though the data contains zero such examples. In particular, all molecules in the data satisfy valency constraints. For example, a carbon atom has four bonds, an oxygen atom has two bonds, and a nitrogen atom has three or five bonds. However, samples from an EBM defined by a neural energy function often violate the valency constraint. While there are methods employing other techniques to enforce valency constraints in the sampling stage (Zang & Wang, 2020), we consider such constraints in a canonical approach to data fitting. Specifically, we express the valency constraint as a statistic function: a valid molecule has zero value from the function, while an invalid molecule has a positive value. Let the constant vector $\mathbf{v} \in \{1, \ldots \nu\}$ store valences for $k$ atom types, with $\nu$ being the maximum valence possible. Then, the valency constraint is:

$$\mathbf{A} \times \mathbf{1} \leq \mathrm{onehot}(\mathbf{b})\mathbf{v}$$

Here $\mathrm{onehot}(\mathbf{b})$ represents each node with a one-hot vector that indicates its atom type, then $\mathrm{onehot}(\mathbf{b})\mathbf{v}$ is a vector indicating valency values of all atoms in the molecule. From this constraint, we define the statistic function as:

$$T(\mathbf{x}) = \mathbf{1}^{\top} \times \max(\mathbf{0}, \mathbf{A} \times \mathbf{1} - \mathrm{onehot}(\mathbf{b})\mathbf{v}), \tag{14}$$

A valid molecule always has $T(\mathbf{x}) = 0$, while an invalid molecule violating the valency constraint has $T(\mathbf{x}) > 0$.

As discussed above, if the model fits the data well, the expected value of $T(\mathbf{x})$ should be 0. Thus, the valency constraint is imposed over samples from the model. The neural energy function may also learn to generate molecules similar to the training data. Our new statistic function $T(\mathbf{x})$ makes the preference explicit in model fitting.

#### 4.2.2  Statistic function for fitting handwritten digits

In the handwritten digits dataset LeCun (1998), we observe that the margin area surrounding the digit has pixels all taking value 0. Fig.1 shows some examples of handwritten digit images, and it can be observed that the pixels in the margin are all zeros. A model with this knowledge will better fit the data. We introduce a statistic function defined as the sum of pixels located at the boundaries of the image. Let $\mathbf{x} \in \mathbf{R}^{h \times h}$ represent one image, where $h$ and $h$ are the height and width of the image, respectively.

Figure 1: All pixels outside the yellow bounding box are zero. This piece of prior knowledge is encoded in the statistics in (15)

Let $\alpha$ be the width of the margin.

$$T(\mathbf{x}) = \frac{\sum_{(i,j) \in S} |\mathbf{x}(i,j)|}{|S|} \tag{15}$$

Here $S$ represents the set of pixels in the margin area $S = \{(i,j)|i \leq \alpha \text{ or } i \geq h - \alpha, \; j \leq \alpha \; \text{ or } j \geq h - \alpha\}$. With this statistic function, the model will explicitly consider pixel values in the margin area during the learning process.

### 4.2.3 Statistic function for fitting point clouds

Point clouds have been widely used in computer graphics, computer vision, and robotics (Achlioptas et al., 2018; Xiao et al., 2023; He et al., 2023; Huang et al., 2024). Several algorithms have been proposed for point cloud generation. In this work, we consider the DPM (Luo & Hu, 2021) and latent diffusion model. These two methods can be taken as directly modeling the joint distribution of all the points, meaning that $\nabla_{\mathbf{x}} F_{\boldsymbol{\theta}}(\mathbf{x})$ is the gradient of all the points' joint distribution in our framework.

For the statistic term, considering smooth surfaces are a characteristic trait of high-quality point clouds, we define the following statistic function for the model:

$$T(\mathbf{x}) = tr(\mathbf{x}^{\top} \mathbf{L} \mathbf{x}). \tag{16}$$

Here $\mathbf{x} \in \mathbf{R}^{N \times 3}$ is the 3D coordinates of N points. $\mathbf{L}$ is the sparse Laplacian matrix of the point clouds' k-nearest neighbor (k-nn) graph. The k-nn graph is constructed using a kd-tree, with a time complexity of $O(n \log n)$ (Preparata & Shamos, 2012), where n represents the number of data points. This is computationally more efficient than the naive k-nn method and can scale better. The function $T(\mathbf{x})$ computes the differences between a node and its neighbors in the graph. It is a measure of the uniformness of points in the point cloud. With this statistic function, the model fitting procedure will explicitly adjust the uniformness of generated point clouds to the same level as training samples.

## 5 Experiments

We evaluate the effect of our special statistics on data fitting through three generative tasks: molecule graph generation, image generation, and point cloud generation. Experiments related to molecule generation are detailed in Section 5.1. Section 5.2 discussed the image generation experiments. Section 5.4 discussed the point cloud generation task. Each section briefly discusses each task's experiment setup and evaluation metrics we used. More details are in the Appendix.

### 5.1 Fitting molecular data

In the first experiment, we incorporate our statistic function derived from valency constraints into an EBM model for molecule generation.

**Experiment Settings:** Our model is evaluated on the QM9 dataset (Pinheiro et al., 2020), which comprises a diverse collection of 133,885 molecules, each containing up to a maximum of 9 atoms. Following previous studies, the molecules are kekulizated using the RDKit library (Bento et al., 2020), and hydrogen atoms are removed. Two essential metrics are employed to evaluate the quality of the generated molecules: validity and valency ratio. A valid molecule must satisfy certain chemical constraints and rules, indicating a chemically reasonable structure. Validity is determined by calculating the proportion of valid molecules without valency correction or edge resampling. We evaluated our strategy on two methods: EDP-GNN(Niu et al., 2020) and GDSS(Jo et al., 2022). We trained GDSS for 300 epochs, the batch size is 1024, and the learning rate is 5e-3. We trained EDP-GNN for 1000 epochs, the batch size is 1024, and the learning rate is 8e-3.

**Results:** We first evaluate the validity ratio of generated molecules. The results are shown in the left column of Tab. 1. Ten thousand molecules are sampled for evaluation. The results show that the model improves the validity ratio of the samples with our new term expressing the validity information. We further check expectation $\mathbb{E}\left[T(x)\right]$ in two distributions with and without the statistic function. Let $\Delta_{T(x)} = \mathbb{E}_{p_{model}}\left[T(x)\right] - \mathbb{E}_{p_{train}}\left[T(x)\right]$ denote the difference between the distribution mean and the sample mean of the statistics term. The results clearly show that our model pays attention to the new term and tries to match its mean to the sample mean of the data. It demonstrates that the extra valency term does help the model learn the valency property.

Table 1: Results for molecule generation on QM9 dataset

| QM9 | Validity ratio(%)↑ | $\Delta_{T(\mathbf{x})}(\downarrow)$ |
|---|---|---|
| EDP-GNN | 88.33 | 1.85 |
| EDP-GNN(with $T(x)$) | **94.52** | **0.98** |
| GDSS | 95.72 | 0.94 |
| GDSS(with $T(x)$) | **96.73** | **0.93** |

Table 2: Negative Likelihood performance on MNIST dataset

| Method | NLL ($\downarrow$) | $\Delta_{T(\mathbf{x})}(\downarrow)$ |
|---|---|---|
| VE-SDE | 3.56 | 6.52 |
| VE-SDE(with $T(x)$) | **3.49** | **6.42** |
| VP-SDE | 3.37 | 6.48 |
| VP-SDE(with $T(x)$) | **3.29** | **6.37** |

## 5.2 Fitting data of hand-written digits

We evaluated the effectiveness of our method on the MNIST dataset, a widely used benchmark for various image downstream tasks. Each image in the MNIST dataset is a 28x28 grayscale representation of a handwritten digit from 0 to 9.

**Experiment Settings:** We experimented on the two forward SDEs: VP-SDE and VE-SDE. The boarding pixels are extracted using a mask. The size of the mask is $22 \times 20$. The model is trained for 1000 epochs and then compared to the likelihood. The batch size is 4096, and the learning rate is 1e-2. The learning rate is kept constant for the first 300 epochs and decreases linearly from 300 to 1000 epochs. The parameter for the statistics term $\eta$ is initialized to be zero. We assess the performances of competing models by comparing their likelihoods on the test set.

**Results:** We evaluated our model via the test set negative log-likelihood (NLL) in terms of bits/dim (bpd). Tab. 2 reports the averaged NLLs of probability flow ODE Song et al. (2020b) over two repeated runs of likelihood computations. The results suggest incorporating the statistics term into the model improves the likelihood. This improvement implies that the model's distribution aligns better with the observed data when considering the inductive bias. The results also show that the difference between the distribution mean and the sample mean of the new statistic is smaller with our method, which implies that our model better captures the data statistic.

## 5.3 Fitting data of small grayscale images

We tested our approach on the FashionMNIST dataset Kayed et al. (2020). Similar to the MNIST dataset, we added the constraint on the boarding pixels. Let $\mathbf{x} \in \mathbf{R}^{h \times h}$ represent one image, where $h$ and $h$ are the height and width of the image, respectively. Let $h$ and $w$ represent the height and width of images in the dataset, and $\alpha$ be the width of the margin, then we use the same statistic function $T(\mathbf{x})$ in (15) that computes the gray level of the border area of an image. In real data, the border area contains mostly white pixels. Without any specification, a typical generative model often overlooks such patterns. Here, we explicitly emphasize this pattern in the model that fits with this statistic function.

**Experiment Settings:** We evaluated performance using the VE-SDE method with a batch size of 64 and an initial learning rate 1e-2. The learning rate decays step-wise by multiplying it by 0.95 every ten epochs. The model was trained for 100 epochs. Boarding pixels were extracted using a $21 \times 19$ mask. We repeated the experiments with different random seeds three times and averaged the results.

**Results:** We assess the performance by comparing the negative log-likelihood (NLL) on the test set and the deviation between the distribution mean and the sample mean of the statistics term ($\Delta_{T(x)}$) against baseline methods. The results, presented in Table 3, indicate our method's performance improvements in NLL and the statistical term difference compared to the VE-SDE without $T(x)$.

Table 3: Negative Likelihood performance on FashionMNIST dataset

| Method | NLL ($\downarrow$) | $\Delta_{T(\mathbf{x})}(\downarrow)$ |
|---|---|---|
| VE-SDE | 4.605 | 47.12 |
| VE-SDE(with $T(x)$) | **4.599** | **43.75** |

Table 4: Comparison of shape generation on ShapeNet. MMD is multiplied by $10^2$. COV is multiplied by $10^2$. Mean diff is the difference between the sample mean and the distribution mean

| Category | Model | MMD ($\times 10^2,\downarrow$) | COV ($\times 10^2,\uparrow$) | 1-NNA (%,$\downarrow$) | $\Delta_{T(\mathbf{x})}$ ($\downarrow$) |
|---|---|---|---|---|---|
| Airplane | DPM | 0.572 | 43.75 | 86.91 | 102.71 |
| | DPM(with $T(x)$) | **0.542** | **45.50** | **85.25** | **91.58** |
| | Latent diffusion model | 0.389 | 49.11 | 68.89 | 39.90 |
| | Latent diffusion model(with $T(x)$) | **0.387** | **49.60** | **67.04** | **37.52** |
| Car | DPM | 1.140 | 34.94 | 79.97 | 326.65 |
| | DPM(with $T(x)$) | **1.137** | **36.83** | **75.19** | **284.98** |
| | Latent diffusion model | 0.802 | 43.70 | 76.23 | 203.02 |
| | Latent diffusion model(with $T(x)$) | **0.781** | **45.31** | **73.3** | **187.90** |

## 5.4 Fitting data of point clouds

Point cloud datasets are typically collections of individual 3D points, each with its position in space and potentially associated attributes, forming a sparse and irregular representation of the underlying scene. Uniformity is a critical factor for generating point clouds. Clumping often occurs when points are concentrated in specific areas, potentially losing detail in other regions. On the other hand, sparsity results in areas with few points, leading to information loss and inaccuracies. Therefore, a generative model needs to pay attention to uniformness among data points to achieve premium results.

**Experiment Settings:** We evaluated our model on the ShapeNet dataset (Chang et al., 2015), a widely used benchmark in 3D shape analysis and understanding. The ShapeNet dataset comprises 51,127 unique objects distributed across 55 categories. Each category represents a distinct class of objects, covering various shapes, such as airplanes, cars, chairs, and animals. The dataset's richness and diversity make it ideal for evaluating generation performance. We evaluated our strategy on the two models. One is the DPM model (Luo & Hu, 2021), and the other is conducted on a latent diffusion model with the decoder is also the score-based model. The latent code is the shape code for each input point cloud. We use PointNet(Qi et al., 2017) to map the input points into a 512-dimension latent feature code for the encoder model. For the decoder and the latent prior score model, we use OccNet (Mescheder et al., 2019), which stacked 6 ResNet blocks with 256 dimensions for every hidden layer. We evaluated our model in two categories: airplane and car. We report Minimum Matching Distance(MMD), Coverage score(COV), and 1-NN classifier accuracy(1-NNA) to evaluate the quality of the samples.

**Results:** The results are shown in Tab. 4. The results demonstrated that integrating smoothness constraints into generating and processing point cloud data improves generation quality. Encouraging a more even distribution of points makes the resulting point cloud representation more accurate, robust, and suitable for various applications. We further evaluate the difference between the sample mean and the distribution mean of the new statistic. We see that our method reduces the gap, which indicates that our model better captures a key statistic in the data.

## 5.5 The quality of the statistic function

One question is whether an arbitrary function can act as a statistic term and improve the data fitting performance. The question is important because if an arbitrary statistic function can improve the model performance, it is easy for a neural model $F(\mathbf{x})$ to pick up such statistics in the energy function. In this section, we set up two types of statistic functions: the first type captures data statistics meaningfully, and the second type has no obvious relationship with the data. For the first type, we still use the statistic

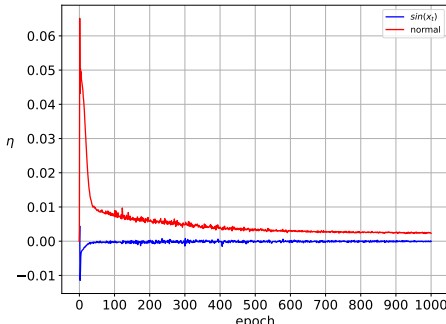

Figure 2: The comparison between $\eta$ values respectively for an arbitrary statistic $\sin(\mathbf{1}^\top \mathbf{x})$ and the mask statistic specified by (15). The result indicates that only specially designed statistics are likely to help the model learn.

function specified by (15). For the second type, we set $T(\mathbf{x}) = \sin(\mathbf{1}^\top \mathbf{x})$, which doesn't seem to capture any reasonable data statistics. We then learn two models respectively with the two statistics and check their corresponding parameters $\eta$. We conduct this experiment on the MNIST dataset.

We get the results in Fig.2, where we plot $\eta$ values against training epochs. We see $\eta$ is positive for the statistic specified in (15). As a comparison, $\eta$ corresponding to the meaningless statistics $\sin(\mathbf{1}^\top \mathbf{x})$ converges to zero, which means that it does not help the model to learn.

Another question arises: how to select a meaningful statistical function for a specific problem to help the model to learn? One approach is to leverage the domain knowledge associated with the problem. For instance, in the context of molecular graph generation, the chemical valency constraint is a critical factor to consider. The domain constraints can be formulated as the statistic function of the generative model to enforce such knowledge.

## 6 Conclusion

In this work, we propose an energy-based model whose energy function includes a neural energy function and a specially designed statistic function. We show that this hybrid model inherits the nice property of exponential-family models, that is, model training approximately matches the statistic's distribution mean to its sample mean. This property allows us to express special statistics in the energy function as an inductive bias. We have shown that this technique can be extensively applied to multiple applications. The experiments show that the proposed strategy improves the modeling performance on three data types, molecular graphs, hand-written digits, and point clouds.

### Acknowledgments

We sincerely thank all reviewers and the editor for their insightful comments. The work was supported by NSF Award 2239869.

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

# A   Appendix

# B   Local minimum

**Theorem.** *If $\boldsymbol{\eta}$ is at a local maximum of the data log-likelihood, then $E_{p_\theta}[\mathbf{T}(x)] = \frac{1}{N} \sum_1^N \mathbf{T}(x_i)$.*

*Proof.* Suppose we have N samples $\{x_1, x_2, \cdots, x_N\}$, and we want to train our model via MLE.

First, the data likelihood can be written as :

$$
\begin{aligned}
l &= \prod_{i=1}^{N} p_{\boldsymbol{\theta},\boldsymbol{\eta}}(\boldsymbol{x}) \\
&= \frac{exp\big[\sum_{i=1}^{N} F_{\boldsymbol{\theta}}(x_i) + \boldsymbol{\eta}^T \sum_{i=1}^{N} \mathbf{T}(\mathbf{x}_i)\big]}{Z^n(\boldsymbol{\theta},\boldsymbol{\eta})}
\end{aligned}
\tag{17}
$$

This leads to the log-likelihood as :

$$
\begin{aligned}
L &= \log \prod_{i=1}^{N} p_{\boldsymbol{\theta},\boldsymbol{\eta}}(\boldsymbol{x}) \\
&= \sum_{i=1}^{N} F_{\boldsymbol{\theta}}(\mathbf{x}_i) + \boldsymbol{\eta}^T \sum_{i=1}^{N} \mathbf{T}(\mathbf{x}_i) - n \log Z(\boldsymbol{\theta},\boldsymbol{\eta})
\end{aligned}
\tag{18}
$$

Then, we take the derivatives on both sides,

$$\nabla_{\boldsymbol{\eta}} L = \sum_{i=1}^{N} \mathbf{T}(\mathbf{x}_i) - \frac{n}{Z(\boldsymbol{\theta}, \boldsymbol{\eta})} \frac{\partial Z(\boldsymbol{\theta}, \boldsymbol{\eta})}{\partial \boldsymbol{\eta}} \tag{19}$$

$$= \sum_{i=1}^{N} \mathbf{T}(\mathbf{x}_i) - \frac{n}{Z(\boldsymbol{\theta}, \boldsymbol{\eta})} \int \mathbf{T}(\boldsymbol{x}) exp(F_{\boldsymbol{\theta}}(\boldsymbol{x}) + \boldsymbol{\eta}^T \mathbf{T}(\boldsymbol{x})) d\boldsymbol{x} \tag{20}$$

$$= \sum_{i=1}^{N} \mathbf{T}(\boldsymbol{x}_i) - n \cdot \mathbb{E}_{p_{\boldsymbol{\theta}}, \boldsymbol{\eta}}(\mathbf{T}(\boldsymbol{x})) \tag{21}$$

If we can find the local minimum, then $\nabla_{\boldsymbol{\eta}} L = 0$, guaranteeing that the sample mean of the property function $\mathbf{T}(\boldsymbol{x})$ equals the expected value even if we added one more constraint. $\qquad \square$

## C   Hyper-parameter sensitivity

We examined the impact of various parameters on the model, with results shown in Table 5 and Table 6.

We examined how the number of neural network layers affects the fitting point cloud data task. In Tab. 5, $\Delta_{T(x)}$ denotes the difference between the distribution mean and the sample mean of the smoothness term. Our model is less sensitive to the hyper-parameter and has a smaller variance than the model without $T(\mathbf{x})$ and has a smaller difference. Similar trends are observed in the task of fitting molecular data. In Tab. 6, $\Delta_{T(x)}$ refers to the difference between the distribution mean and the sample mean of the valency term. The results show that as the number of GNN layers increases, EDPGNN exhibits a higher variance in the validity ratio and $\Delta_{T(\mathbf{x})}$, whereas EDPGNN with $T(\mathbf{x})$ shows minor changes.

Table 5: Effect of number of neural network layers

| Number of neural network layers | Latent diffusion model $\Delta_{T(\mathbf{x})}$ | Latent diffusion model (with $T(x)$) $\Delta_{T(\mathbf{x})}$ |
|:---:|:---:|:---:|
| 2 | 45 | 40.14 |
| 4 | 42 | 39 |
| 6 | 39.9 | 37.52 |

Table 6: Effect of the number of GNN layers

| Number of GNN layers | EDPGNN Validity ratio (%) | EDPGNN (with $T(\mathbf{x})$) Validity ratio (%) | EDPGNN $\Delta_{T(\mathbf{x})}$ | EDPGNN (with $T(\mathbf{x})$) $\Delta_{T(\mathbf{x})}$ |
|:---:|:---:|:---:|:---:|:---:|
| 1 | 0.863 | 0.933 | 1.98 | 1.05 |
| 2 | 0.878 | 0.938 | 1.91 | 1.02 |
| 3 | 0.882 | 0.945 | 1.85 | 0.99 |
| 4 | 0.883 | 0.946 | 1.87 | 0.98 |

