# OpenReview forum: "Incorporating Inductive Biases to Energy-based Generative Models"
_TMLR — Accepted by TMLR_

### Review · Reviewer_s9Tb · 2024-06-13

**Summary Of Contributions:**

This work introduces a hybrid approach that combines energy-based models with exponential family models to incorporate inductive bias in data modeling. By augmenting the energy function with a well-studied statistic function, the model captures relevant specified rules associated with the statistic function, providing inductive bias. Empirical studies confirm the effectiveness of the statistic-matching property when informative statistics are utilized.

**Audience:**

Yes

**Claims And Evidence:**

Yes

**Requested Changes:**

**Improvement suggestions for theoretical explanation:**
- Enhance the explanation of the difference between the distribution mean and the data mean by utilizing Equation 9 and discussing its relation to $\Delta_{T(x)}$ (where absolute values are taken).
- Provide a clearer and more precise discussion on the convexity of the negative likelihood.

**Typo corrections:**
- *Abstract:* Specifically, "we" augment the energy term.
- *Page 3:* A neural network with learnable parameters $\theta$.
- Change uppercase $L$ to lowercase$l$ in the statement of Theorem 1 as well as its proof.
- Address the missing reference for "[cite]" where methods employing other techniques to enforce valency constraints are discussed.
- The usage of the symbol $*$ in Equation 8 is not common, as it is typically associated with convolution.

**Strengths And Weaknesses:**

**Strengths**
- The idea of this paper is simple yet effective: the proposed approach directly combines two existing methods, successfully incorporating known knowledge through the statistic function into the energy-based model.
- The paper offers theoretical evidence for the proposed method, affirming that the hybrid approach maintains the advantages of utilizing the statistic function.
- The paper includes some examples of statistic functions to illustrate the broad applicability of their approach.

**Weakness**
- I suggest that the explanation of theoretical results could be further improved by providing more specific details or citing related references to make it easier for a general audience to understand.

---

> ### Author Response · Authors · 2024-08-01
> **About the discussion of the theoretical results**
>
> Thank you for your positive feedback. We have fixed all these typos in the updated version.
>
> **Q1:** *Enhance the explanation of the difference between the distribution mean and the data mean*
>
> The updated version has enhanced the explanation of the difference between the distribution mean and the data mean and cited related references. It also gives more precise citations. We have modified the discussion on the convexity of the negative likelihood. The convexity of the negative likelihood can be proved by the convexity of the log partition function [Wainwright et al., 2008].  We have also removed the operation of taking the absolute value as the difference itself approaches zero. We hope that our revisions and clarifications address your concerns.
>
> Reference:
>
> Wainwright, Martin J., and Michael I. Jordan. "Graphical models, exponential families, and variational inference." Foundations and Trends® in Machine Learning 1.1–2 (2008): 1-305

---

### Review · Reviewer_i2Kx · 2024-07-20

**Summary Of Contributions:**

The given work proposes an energy based model which incorporates a neural energy function and novel parameter less statistic function.  The authors argue that this approach inherits property of exponential family models which allows them to express special statistics  in energy function as an inductive bias. The authors also present an empirical study on three tasks (molecule graph generation, image generation, point cloud generation)  to verify their claims of statistic matching .

**Audience:**

Yes

**Broader Impact Concerns:**

Not applicable.

**Claims And Evidence:**

Yes

**Requested Changes:**

Major points:
* Clarity of writing around mathematical formulation and individual statistical functions for the empirical evaluation.
* Hyper-parameter sensitivity and  empirical evaluation for higher dimensional datasets.


Minor point:
 * Few grammatical corrections. For example "Specifically, We augment" -> "Specifically, we augment"  in line 5 of the abstract.

**Strengths And Weaknesses:**

Strength:

* Novel theoretical contribution on the hybrid model which combines energy based models with exponential family models.
* Theoretically well grounded with past work clearly cited.

Weakness:

* Clarity of writing: Specifically some of the mathematical formulations in Section 4 which are hard to follow, especially for those not as well versed in this specific domain. Also in the subsections of 4.2 for each of the empirical statistic functions.

* Ablation studies and scalability: While the early results are encouraging, it would be interesting to see the results on high dimensional widely adopted benchmarks to give s sense of scalability. Additionally, while this is partially addressed in  5.4 (Fig 1), it would be interesting to see the influence of various subcomponents of this hybrid model to the overall metrics.

---

> ### Author Response · Authors · 2024-08-01
> **About the clarity of the work and ablation studies**
>
> **Q1:** *Clarity of writing*
>
> Thank you for pointing out our writing issues. We have revised section 4.2.1 to better describe the task of fitting molecular data. We have also revised the notation to improve the clarity of the formulation of valency constraints. We have also added a figure to illustrate the statistics we use for modeling MNIST digits. We hope Section 4 is easier to understand now.
>
> **Q2:** *Ablation studies and hyper-parameter sensitivity*
>
> Our new method does not have multiple components, so we either include the statistics or not in the model. The comparison has been done in our experiments. Please advise if you have a different idea of ablation studies. We agree that it is useful to test the method against baselines with different hyperparameters and architectures. We are running the experiment now and will report back soon.
>
> **Q3:** *Empirical evaluation for higher dimensional datasets*
>
> We are looking for a high-dimensional dataset that fit our evaluation purpose. We also would like to point out that the point cloud data has 10K dimensions, which can validate the effectiveness of the proposed approach on high-dimension conditions. The scalability of the new method is not an issue, though we don't have much discussion about it. A diffusion-based training method does require intensive computation on high-dimensional data. However, our method only increases computation by taking the gradient of the statistic function with respect to the random variable. The increased computation is insignificant since the statistic function is much simpler than the neural network function.

---

### Review · Reviewer_HrAQ · 2024-07-21

**Summary Of Contributions:**

- This paper proposes a hybrid energy function that combines a neural energy function with a specially designed statistic function. This design choice allow us to utilize an inductive bias for each task.
- This paper demonstrates the effectiveness of the proposed method in three applications: molecule generation, image generation, point cloud generation.

**Audience:**

Yes

**Claims And Evidence:**

No

**Requested Changes:**

Referring to Weaknesses.

**Strengths And Weaknesses:**

Although this paper is generally well-written and easy to follow, but I have several concerns on this paper and I cannot find enough strengths to accept this paper for this journal. I have described my concerns below.

1. **Limited methodological contribution**
    - This paper simply adds $\eta^\top\mathbf{T}(\mathbf{x})$ to the energy function, where $\mathbf{T}(\mathbf{x})$ is manually-designed statistic function based on inductive bias. In many cases, it is hard to design the function explicitly, which is why we use/learn a neural energy function. I think the proposed approach is not broadly applicable.
    - As mentioned by the authors, there might be other technique to enforce a specific constraint (e.g., in molecule generation, atom-by-atom sequential generation can avoid cases where the valency constraint is violated). I'm not convinced why and when $\mathbf{T}(\mathbf{x})$ is superior to other techniques.
    - Some functions cannot be (or are hard to be) applied to the intermediate sampling process: e.g., EBM sampling via Langevin dynamics and diffusion process. For example, the valency constraints in molecule generation cannot be utilized before atom's characteristic is determined (eq 14), and smoothness of the surface of $N$ points (eq 16) requires $O(N^2)$ operations to construct k-nn graph of the point cloud.
    - In image generation, why is the statistic function (eq 15) required? I think the function limits the quality of generated samples. Can this function be generally applied in image generation?
2. **Experimental results are not convincing**
    - This paper conducts experiments with only a small set of benchmarks. In particular, MNIST benchmark is not enough to demonstrate the effectiveness of the proposed method in image generation tasks.
    - The proposed method is not compared with a sufficient number of baselines. There are many generation frameworks for molecule generation, image generation, and point cloud generation. Therefore, I cannot be convinced that the proposed method improves the generation quality by a large margin.

---

> ### Author Response · Authors · 2024-08-01
> **About the applicability and generation performance of the new approach**
>
> **Q1:** * The proposed approach is not broadly applicable.*
>
> The proposed approach aims to provide a new method of fitting energy-based models. Our focus is on a general problem of data fitting, a fundamental problem in statistical machine learning. The proposed method is particularly suitable for applications where prior information can be incorporated into the energy function. We argue that practitioners often have insights to provide such prior. As mentioned by the renowned book “Bayesian Data Analysis” [Gelman et al., 2003], data fitting is an iterative process. A practitioner may observe a discrepancy between the data and the fitted model. The discrepancy is often expressed as the difference between statistics from model samples and real data. Therefore, it is not hard to find and define statistics in practice. Our method provides a generic tool for practitioners to address such discrepancies. We have revised the introduction section to include more discussions there.
>
> **Q2:** *There might be other technique to enforce a specific constraint (e.g., in molecule generation, atom-by-atom sequential generation can avoid cases where the valency constraint is violated)*
>
> We’d like to point out that the focus of the work is not generation but data fitting. Many generative models enforce generation constraints during the generation/sampling stage, so these techniques do not change the model fitting procedure. Our work has a different aim: we want to fit a better model so that it learns known data statistics/constraints. We believe improving data fitting is a valuable contribution to the development of energy-based models.
>
> **Q3:** *Some functions cannot be (or are hard to be) applied to the intermediate sampling process... For example, the valency constraints in molecule generation cannot be utilized before atom's characteristic is determined (eq 14), and smoothness of the surface of 𝑁 points (eq 16)*
>
> We have improved our writing in 4.2.1 to clarify how to incorporate valency constraints. We can express the constraint as a term as a relatively simple function (eq 14) in the energy function. The sampling procedure will try to satisfy the constraint in Langevin dynamics, which means that it will consider both atom types and bond types at the same time to satisfy the constraint.
>
> In the smoothness constraint, we do have some approximation that we don’t consider the neighbor graph as a function of $x$. This is acceptable because, in the later stage of the sampling procedure, the neighbor graph does not have a lot of changes. At the same time, the improvement of smoothness happens mainly at this stage because of our smooth term.
>
> **Q4:**  *In image generation, why is the statistic function (eq 15) required? I think the function limits the quality of generated samples. Can this function be generally applied in image generation?*
>
> We have updated the manuscript to include an illustrative example in Section 4.2.2. The examples demonstrate that the pixels in the margin areas have value zeros. We want the model to capture this pattern to improve the data fitting. For specialized image generation tasks (e.g. generating images of cells), there are still statistics (e.g. whether the generated cell matches a particular template) to guide the model for better generation results. For general image generation, there are statistics for low-level image features (e.g. smoothness). For high-level features (whether a hand has 5 fingers), one may need to train a separate neural network and fix it as the statistic function. But this is beyond the discussion of this work.
>
> **Q5:** *This paper conducts experiments with only a small set of benchmarks. In particular, MNIST benchmark is not enough to demonstrate the effectiveness of the proposed method in image generation tasks.*
>
> We have revised our section title to "Fitting data of hand-written digits" to better reflect the scope of our experiment. We include three different tasks in our experiment section to demonstrate that the proposed approach is applicable across various domains.
>
> **Q6:** *The proposed method is not compared with a sufficient number of baselines. There are many generation frameworks for molecule generation, image generation, and point cloud generation. *
>
> We would like to point out that the main purpose is to improve *data fitting* with *energy-based models*. Generation performance is certainly one evaluation metric of data fitting but is not the sole focus of the work. We also evaluate our new method with negative log-likelihoods and discrepancy of key statistics to evaluate different models. While recent models designed for generation tasks (e.g. flow matching)  have achieved good generation performance, they are not energy-based models and do not have an approach to target a particular data statistic in data fitting. A comparison with such a model may not provide insights about how to improve data fitting with energy-based models.

---

### Author Response · Authors · 2024-07-22
**Thanks for all reviewers!**

We will post our responses shortly!

-- Authors

---

> ### Author Response · Authors · 2024-08-01
> **Changes to the submission**
>
> Thank reviewers again for your feedback. There seems to be an agreement that the proposed approach is new and effective, but there are also concerns about the applicability of the model and its generation performances. We have made the following changes to address your concerns.
>
> 1. We have strengthened the discussion of how the new approach provides a tool for practitioners to improve energy-based models.
> 2. We have clarified the derivation of the gradient calculation and provided better citations.
> 3. We have revised the description of developing statistics for the three tasks.
> 4. We also have changed section titles in the experiment section to accurately reflect the scope of each experiment.
>
> Below we address your concerns separately.

---

### Decision · Action_Editor_HHD1 · 2024-09-28

**Recommendation:** Accept as is

**Comment:**

The authors introduce a simple yet effective approach that incorporates inductive bias into energy-based models (EBMs) for data fitting. They demonstrate the effectiveness of the approach across three different domains: images, point clouds, and molecules. While one reviewer remains negative after the discussion, the majority of the reviewers, along with Action Editor (AE), agree that the claims made in the submission are well-supported by clear evidence. AE also believes that practitioners would find this approach valuable, as inductive bias is often known by domain experts in real-world applications, making the approach both applicable and capable of improving EBM performance.

Therefore, AE recommends acceptance but strongly encourages the authors to conduct additional experiments to further validate the effectiveness of the proposed approach.

**Audience:**

Yes, practitioners may find this approach valuable, especially in some applications where inductive bias is known by domain experts.

**Claims And Evidence:**

Yes, the claims made by authors have been supported by clear evidence.

---

> ### Author Response · Authors · 2024-10-08
>
> Thank you for reviewing our work! We will include additional experiments and revise our manuscript in the camera-ready version.

---

> > ### Author Response · Authors · 2024-10-31
> > **Camera-ready version**
> >
> > Dear Editor and Reviewers,
> >
> > We have done another editing pass and fixed a few small typos. We have also added another experiment with the FasionMNIST dataset. We still observed the performance improvement from our method.
> >
> > Thank you again for your valuable feedback!
> >
> > --Authors